# Formation of marine atmospheric organic aerosols associated with the spring phytoplankton bloom after sea ice retreat in the Sea of Okhotsk

Yuzo Miyazaki<sup>1</sup>, Yunhan Wang<sup>1,2</sup>, Eri Tachibana<sup>1</sup>, Koji Suzuki<sup>2,3</sup>, Youhei Yamashita<sup>2,3</sup>, Jun Nishioka<sup>1</sup>

- <sup>1</sup> Institute of Low Temperature Science, Hokkaido University, Sapporo, 060-0819, Japan
- <sup>2</sup> Graduate School of Environmental Science, Hokkaido University, Sapporo, 060-0810, Japan
- <sup>3</sup> Faculty of Environmental Earth Science, Hokkaido University, Sapporo, 060-0810, Japan
- 10 Correspondence to: Yuzo Miyazaki (yuzom@lowtem.hokudai.ac.jp)

Abstract. The Sea of Okhotsk is one of the most biologically productive regions, where primary production during spring phytoplankton blooms after sea ice melting/retreat has the potential to contribute to the sea-to-air emission flux of atmospheric organic aerosols (OAs). To elucidate the effect of oceanic biological activity during blooms on the formation process of OAs, aerosol samples and surface seawater were collected during the bloom period of April 2021. Organic matter (OM) was the dominant component of submicrometer aerosols during both the bloom (53±16%) and bloom-decay periods (44±12%), with OM being highly water-soluble during the bloom. Stable carbon isotope ratios of aerosol organic carbon (OC) showed that 73–82% of the observed aerosols were of marine origin. Relations between water-soluble OC (WSOC) and molecular tracers suggested that aerosol WSOC of marine origin was likely affected by secondary formation from precursors such as α-pinene and DMS-relevant compounds rather than primary emissions of sea spray aerosols. The amounts of water-soluble organic nitrogen (WSON) in aerosol and dissolved organic nitrogen (DON) in seawater during the bloom were larger than those during bloom-decay period, suggesting the preferential formation of N-containing water-soluble OAs of marine origin during the bloom. The increase in the amount of DON during the bloom was likely associated with the predominant diatoms, *Thalassiosira* spp. and *Fragilariopsis* spp. This study highlights the significant contribution of the secondary formation of marine biogenic OAs with increased N-containing components during the bloom after sea ice melting/retreat in the subarctic ocean.

## 1 Introduction

Ocean surfaces are a significant source of atmospheric organic aerosols (OAs), which are formed by direct emissions of particles (i.e., sea spray aerosols) or by volatile organic compounds (VOCs) followed by particle formation (i.e., secondary OAs). These ocean-derived OAs affect the activities of cloud condensation nuclei (CCN)

and ice nucleating (IN) particles and thus play a vital role in climate systems by controlling the atmospheric radiative budget (Cochran et al., 2017). In particular, the chemical composition, concentration, and size distribution of these ocean-derived OAs controls their ability to influence climate. The Sea of Okhotsk is one of the most productive marginal seas in the global oceans because of favorable light conditions and high inputs of macronutrients and iron from regions such as the Amur River (Nishioka et al., 2007; Liu et al., 2009; Sohrin et al., 2014; Nishioka et al., 2021).
The Sea of Okhotsk plays an important role in supplying organic matter (OM) and iron to the western subarctic Pacific, which is characterized by seasonal sea ice (Kimura and Wakatsuchi, 2000; Yamashita et al., 2021). After the ice melts in spring, the Sea of Okhotsk hosts large phytoplankton blooms (Sorokin, 1999; Mustapha et al., 2009; Shiozaki et al., 2014; Kishi et al., 2021).

Marine biogeochemical cycles at high latitudes are driven by marine microalgae in seawater and sea ice through the fixation and export of carbon, uptake of nutrients, and the production and release of oxygen and organic compounds (Young et al., 2020). Ice algae live in and under sea ice and play a principal role in primary production in sea ice areas (Syvertsen, 1991). As snow melts, light penetrates deeper into ice-covered water, triggering substantial phytoplankton blooms (Arrigo et al., 2012; Ardyna et al., 2020). In the Sea of Okhotsk, ice algal diatoms constitute the majority of ice algae (Nakamura et al., 2020; Watanabe, 2022) and often dominate the primary production and phytoplankton biomass during blooms in high-latitude marine environments (Brzezinski et al., 2001; Assmy et al., 2013; Yan et al., 2020; 2022).

Several biological particles produced by diatoms on the sea surface can increase the IN temperature of small water droplets and act as IN particles (Irish et al., 2017). For example, Eickhoff et al. (2023) suggested that the sea ice diatom *Fragilariopsis cylindrus* and its fragments may contribute to IN in the marine environments of polar regions. Some of these biological particles can increase the IN temperature of small water droplets and act as ice-nucleating particles (Ickes et al., 2020). These biological particles can be transported to the atmospheric boundary layer by sea spray aerosols (Irish et al., 2019; Steinke et al., 2022), and/or their exudates may be associated with the formation of secondary organic aerosols (SOAs) over the sea surface.

Oceanic biological activity during phytoplankton blooms associated with sea ice melting is expected to act as a significant source of atmospheric OAs and has the potential to change the subsequent physicochemical characteristics of aerosols, affecting the atmospheric environment through IN and CCN. To date, no data on atmospheric OAs associated with spring blooms in the Sea of Okhotsk are available. Understanding the origin and formation processes of marine OAs related to marine microbial activity is essential, particularly those associated with phytoplankton blooms in the subarctic oceans. However, the amount and relative contributions of primary and secondary formations to OAs in the subarctic oceans remain uncertain. The purpose of this study was to elucidate how oceanic biological activity during blooms in the Sea of Okhotsk affects the amount and formation process of atmospheric OAs, which

were investigated using atmospheric aerosol and surface seawater samples obtained by shipboard measurements in the southern Sea of Okhotsk in the spring of 2021.

## 65 2 Methods

85

## 2.1 Field observations

Ambient aerosol and surface seawater samples were collected in the southern Sea of Okhotsk during the KS-21-6 expedition of the R/V *Shinsei Maru* (JAMSTEC) in April 2021. During this period when the sea ice had already melted or retreated (**Fig. S1**), a phytoplankton bloom was observed in the Sea of Okhotsk and the surrounding oceanic area (**Fig. 1a**). The track of the R/V *Shinsei Maru* is also shown in **Fig. 1b** and **1c**.

## 2.2 Ambient aerosol sample collection

Ambient submicrometer aerosol samples were collected continuously from April 6 to April 25, 2021, using a high-volume air sampler (HVAS; Model 120SL, Kimoto Electric, Osaka, Japan) on the deck above the bridge of the ship. A cascade impactor (CI; Model TE-234, Tisch Environmental, Cleves, OH, USA) attached to the HVAS was used to collect size-segregated particles at a flow rate of  $1130 \text{ L min}^{-1}$  without temperature and humidity control. In this study, we focused on the analytical results obtained from the bottom stage of the impactor, which collected particles with an aerodynamic diameter ( $D_p$ ) of < 0.95  $\mu$ m and are referred to as submicrometer particles. Aerosol sampling was conducted during the local daytime and nighttime, with a sampling duration of approximately 12–24 h. The samples were collected on quartz fiber filters (25 cm × 20 cm), which were pre-combusted at 410°C for 6 h to remove any contaminants. Four field blanks were collected with quartz-fiber filters mounted on the impactor without running the HVAS, which were obtained on the ship during the expedition. The filter punch was located in glass and plastic vials for the analysis of organic and inorganic species, respectively. All the vials were cleaned with detergent and then washed with distilled water. The glass vials were then baked at 410 °C for two hours to remove contaminants prior to the extraction process.

Nine-stage size-segregated aerosol samples were also collected using an Andersen cascade impactor located next to the HVAS at a flow rate of 120~L min $^{-1}$ . The aerosol particles were collected on precombusted 80-mm-diameter quartz filters based on the 50% equivalent aerodynamic cutoff diameters between 0.39 and 10.0  $\mu$ m (i.e.,  $D_p < 0.39\mu$ m,  $D_p = 0.39-0.58~\mu$ m, 0.58–1.0  $\mu$ m, 1.0–1.9  $\mu$ m, 1.9–3.0  $\mu$ m, 3.0–4.3  $\mu$ m, 4.3–6.4  $\mu$ m, 6.4–10.0  $\mu$ m, and  $D_p > 10.0~\mu$ m). Aerosol sampling using this cascade impactor was performed for approximately 72 h per sample without temperature

or humidity control. All the filter samples collected by the two samplers were individually stored in glass jars with a Teflon-lined screwed cap at -20 °C to limit chemical reactions on the filter and losses of volatile compounds.

Possible contamination from the ship exhaust was avoided by shutting off the sampling pump when air came from the beam and/or when the relative wind speed was low (< 3 m s<sup>-1</sup>). In this study, the results of submicrometer particles collected by the HVAS are mainly shown, whereas the results of aerosol particles collected by the cascade impactor are presented only to show the size distributions of water-soluble fractions of organic matter (section 3.3).

## 2.3 Collection of bulk surface seawater samples

Surface seawater (SSW) was collected during aerosol sampling. The sampling location covered areas moved by the R/V *Shinsei Maru* (**Fig. 1b** and **1c**). The SSW samples used in this study were collected at the sea surface (0 m) and 5-m depth at each station using an acid-washed bucket and a conductivity, temperature, and depth (CTD) instrument equipped with an acid-washed Niskin-X bottle. The bucket was precleaned with 10% (v/v) hydrochloric acid and then washed with Milli-Q water. A towed-fish continuous surface clean sampling (Tsumune et al. 2005) was also conducted to collect SSW during the cruise. The towed-fish was set to a position ~5 m away from the starboard side of the vessel, and seawater was pumped up using a Teflon Bellows pump (Astipure, Ltd.) from the surface layer (1.5–3 m depth) to a clean air booth in the shipboard laboratory.

## 2.4 Chemical analysis of aerosol samples

## 2.4.1 Water-soluble organic carbon (WSOC) and water-soluble total nitrogen (WSTN)

In this study, aerosols extracted with ultrapure water, followed by filtration through a syringe filter, were defined as water-soluble aerosols. To determine the water-soluble organic carbon (WSOC) and water-soluble total nitrogen (WSTN), a filter area of 7.07 cm² and 3.13 cm² was cut from the quartz fiber filters of each submicrometer sample (HVAS) and sample obtained by the Andersen sampler, respectively. The filter cut was extracted using an ultrasonic bath for 15 min with ultrapure water of 15 ml. The total extracts were then filtrated into a combustion tube with a disc filter (Millex-GV, 0.22 μm, Millipore, Billerica, MA, USA) followed by injection of the dissolved sample into a total organic carbon (TOC) analyzer with a total nitrogen (TN) unit (Model TOC-L<sub>CHP</sub>+TNM-<sub>L</sub>, Shimadzu) (Miyazaki et al., 2020). Next, the injection was combusted to 720°C to derive CO<sub>2</sub> and nitrogen monoxide under a constant flow of ultrapure air. After cooling and dehydration with an electrical dehumidifier, the combustion gases were measured to determine the derived CO<sub>2</sub> as TOC by a nondispersive infrared (NDIR) detector. The measured TOC was defined as water-soluble organic carbon (WSOC). TN was converted to nitrogen monoxide, measured with an ozone chemiluminescence detector, and defined as WSTN in this study. Before use, the glassware used for extraction was

washed with ultrapure water, dried in a dryer for 3 h, and heated in a high-temperature chamber at 450°C for 3 h. The blank value accounted for the mass of <30% of WSOC and WSTN. The mass concentrations of WSOC and WSTN were calculated by subtracting the blank values.

## 2.4.2 Inorganic ions

Another cut of filters for each submicrometer aerosol sample (28.3 cm²) as well as samples obtained by the Andersen impactor (3.14 cm²) was extracted with ultrapure water under ultrasonication to determine the concentration of major anions and cations (SO<sub>4</sub><sup>2-</sup>, NO<sub>3</sub><sup>-</sup>, NO<sub>2</sub><sup>-</sup>, Cl<sup>-</sup>, Br<sup>-</sup>, NH<sub>4</sub><sup>+</sup>, Na<sup>+</sup>, K<sup>+</sup>, Ca<sup>2+</sup>, Mg<sup>2+</sup>) as well as methanesulfonic acid (MSA) using an ion chromatograph (model 761 compact IC; Metrohm, Herisau, Switzerland) (Miyazaki et al., 2016). The blank value accounted for the mass of <27% of inorganic ions. The mass concentrations of inorganic ions were calculated by subtracting the blank values. The concentration of inorganic nitrogen (Inorg-N) was defined as the sum of those of nitrate (NO<sub>3</sub><sup>-</sup>), nitrite (NO<sub>2</sub><sup>-</sup>), and ammonium (NH<sub>4</sub><sup>+</sup>) nitrogen measured with IC. The WSON concentration was determined by subtracting the inorganic nitrogen concentration from the WSTN concentration (WSON = WSTN – Inorg-N).

# 2.4.3 Organic carbon (OC) and elemental carbon (EC)

Mass concentrations of organic carbon (OC) and elemental carbon (EC) were determined with a 1.54-cm<sup>2</sup> punch of a submicrometer filter sample using a carbon analyzer (Sunset Laboratory, Inc., Tigard, Oregon, USA). Both OC and EC on the filter sample were thermally converted to carbon dioxide (CO<sub>2</sub>), which was subsequently measured using NDIR.

Water-insoluble organic carbon (WIOC) is defined as [WIOC] = [OC] – [WSOC]. The mass concentrations of water-soluble organic matter (WSOM) and water-insoluble organic matter (WIOM) were calculated by assuming a factor of 1.8 for the conversion of the mass of WSOC to that of WSOM (Finessi et al., 2012) and a factor of 1.2 to convert the mass of WIOC to that of WIOM (Yttri et al., 2007).

#### 2.4.4 Stable carbon isotopic characterization of aerosols

For the determination of stable carbon isotope ratio ( $\delta^{13}$ C) of total carbon (TC = OC + EC) ( $\delta^{13}$ C<sub>TC</sub>), a filter cut of the submicrometer aerosol sample was put in a pre-cleaned tin cup. The  $\delta^{13}$ C<sub>TC</sub> value was then measured using an elemental analyzer (EA) (Flash EA 1112) continuous flow carrier gas system (ConFlo)) interfaced to an isotope ratio mass spectrometer (Finnigan MAT Delta V, Thermo Finnigan, San Jose, CA, USA).

For all samples obtained in this study, the mass concentrations of TC were mostly OC (ave. 92%). Therefore, OC concentration can be regarded to be nearly equal to that of TC (TC  $\approx$  OC) and the  $\delta^{13}C_{TC}$  value was assumed to be equal to the  $\delta^{13}C_{OC}$  value (i.e.,  $\delta^{13}C_{TC} \approx \delta^{13}C_{OC}$ ) in this study.

# 2.4.5 Molecular tracers of organic compounds

The concentrations of the biogenic organic aerosol tracers were measured using the following method. Tracer compounds include oxidation products of α-pinene (i.e., 3-methyl-1,2,3-butanetricarboxylic acid (3-MBTCA)) (Szmigielski et al., 2007), and those of isoprene (2-methyltetrols: the sum of 2-methylerythritol and 2-methylthreitol) (Claeys et al., 2004). A portion of the submicrometer aerosol sample filter (28.274 cm²) was extracted with 5 mL of a mixture of dichloromethane (DCM) and methanol (MeOH) (2:1) for 10 min using an ultrasonic cleaner (output: 90 W) (Fu et al., 2009). The particulate impurities were removed by passing the extract through a Pasteur pipette filled with quartz wool, which was repeated three times. The extract was then concentrated to approximately 1 mL using a rotary evaporator and placed in a 5 mL vial. The rotary evaporator was run at 25°C, and the gauge on the vacuum pump read 80 mmHg.

A mixture of MeOH and DCM was used to recover as much of the sample as adhered to the pear-shaped flask. After evaporating MeOH and DCM in a vial with a stream of nitrogen, a mixture of N, O-bis-(trimethylsilvl) trifluoroacetamide (BSTFA) and pyridinic acid (5:1) was added. The functional groups -COOH and -OH in the extracts reacted with BSTFA to form trimethylsilyl (TMS) esters and TMS ethers, respectively (Fu et al., 2009). After the derivatization, the derivative was diluted with hexane containing the internal standard (n-Tridecane (C<sub>13</sub>) with concentration of 1.43 µg ul<sup>-1</sup> in hexane). Two µL of the TMS derivative was then injected into a capillary gas chromatograph (GC8890, Agilent) equipped with a DB-5MS fused silica capillary column (30 m × 0.25 mm i.d., 0.25 um film thickness) coupled to a mass spectrometer (MSD5977B, Agilent) to determine the concentration of each molecular tracer. The mass spectrometer was operated in the electron ionization (EI) mode at 70 eV. The sample injection was made in splitless mode. The peaks of the target compounds in total ion chromatograms (TICs) were identified by comparison of mass spectra with those of authentic standards or literature data. 3-MBTCA was estimated using the response factor of pimelic acid, which was determined using an authentic standard. 2-methyltetrols was quantified using the response factor of meso-erythritol (Fu et al., 2009). The mass concentrations of molecular tracers were determined by the MS peak area of TMS derivative relative to that of the 140 µl internal standard injected into the GC-MS. Recoveries of each organic compound were measured using the surrogates that were spiked into precombusted quartz-fiber filters (n=3), which were higher than 81% for all the compounds measured. Reproducibility of the measurements is based on relative standard deviation of the concentrations based on duplicate analysis, which was generally <13%. The blank value accounted for the mass of <3% of the organic compounds. The mass concentrations of organic compounds were calculated by subtracting the blank values.

#### 2.4.6 Excitation fluorescence characterization

Fluorescence characteristics were obtained for the water-soluble submicrometer aerosol samples. Briefly, water-extracted aerosol samples obtained using the same procedure as for WSOC were used to measure the excitation-emission matrix (EEM) fluorescence using a fluorometer (FluoroMax-4, Horiba) (Mizuno et al., 2018). All fluorescence spectra were acquired in S/R mode with instrumental bias correction. Subsequently, the EEM of Milli-Q water was subtracted from the sample EEMs. Finally, each EEM was calibrated to the water Raman signal, and the fluorescence was reported in Raman units (RU; nm<sup>-1</sup>). Based on the ranges reported by Coble (2007), the fluorescence intensity of peak T (protein-like; Ex/Em = 275/330 nm) was obtained from the excitation/emission pairs.

## 190 2.5 Measurements of chemical and biological parameters in seawater

Chlorophyll *a* (Chl *a*) concentration in the SSW sample was measured using 0.5 L of SSW sample, which was placed in a black bottle, followed by filtering through a 20-µm pore size, 47-mm diameter nylon mesh with a 2-µm pore size, 47-mm diameter GVS PS membrane filter using a hand pump. Suction filtration was then performed using a 0.7 µm pore size, 25 mm diameter Whatman GF/F filter under a gentle vacuum (< 0.013 MPa) to measure Chl *a* from phytoplankton in different sizes. The filters were soaked in *N*, *N*-dimethylformamide (DMF) at -20 °C for at least 24 hours (Suzuki and Ishimaru, 1990). The concentrations of the Chl *a* extract in DMF were analyzed using the non-acidic method with a Turner Designs 10-AU Field Fluorometer (Welschmeyer, 1994). In this study, Chl *a* concentration refers to the sum of each size of Chl *a* concentrations.

Dissolved organic carbon (DOC) and total dissolved nitrogen (TDN) concentrations in the SSW were measured using the same TOC/TN analyzer used for the aerosol samples. The SSW samples were filtered with a 0.22-µm Durapore filter (Millipore) under a gentle vacuum followed by transferring into a pre-combusted borosilicate glass vial with an acid-cleaned Teflon-lined cap (Mizuno et al., 2018). The samples were then kept frozen at -20 °C in the dark until analysis.

Nutrients (NH<sub>4</sub><sup>+</sup> and NO<sub>3</sub><sup>-</sup>+NO<sub>2</sub><sup>-</sup>) in the SSW samples were measured by an auto-analyzer (QuAAtro; BL tec, Inc) with a continuous flow system (Nishioka et al., 2020). A quality check was performed using a reference seawater material (Kanso Technos). A calibration certificate was obtained from the nutrient-free seawater based calibration curve, as well as the seawater standard. The concentrations were measured using mean values and standard deviations,

which were in good agreement with the certified values. Dissolved organic nitrogen (DON) concentration was calculated by subtracting the concentration of inorganic nitrogen (i.e.,  $NH_4^+ + NO_3^- + NO_2^-$ ) from that of TDN.

# 2.6 Air mass back trajectory

Air mass history was examined by use of the HYSPLIT trajectory model (https://www.ready.noaa.gov/HYSPLIT\_traj.php). Air mass back-trajectories of 48-hour length were calculated arriving at 50 m above ground level of the sampling location for every hour. Trajectories were calculated based on meteorological files from the NCEP/NCAR Reanalysis Data which has a resolution of 1.0° latitude/longitude.

## 3 Results and discussion

## 3.1 Mass concentrations and chemical fractions of submicrometer aerosols in the Sea of Okhotsk

As shown in **Fig. 2**, the average Chl *a* in SSW during the first half of the sailing (April 13–20) was 5.58±2.05 mg m<sup>-3</sup>, which was much larger than that 4.00±1.77 mg m<sup>-3</sup> during the second half of the period (April 24–27). In this study, the first half of the sailing period was defined as the bloom period, and the latter half was defined as the bloom-decay period. **Fig. 3a** shows the time series of chemical mass concentrations of each submicrometer aerosol component during the sailing. OM dominated the submicrometer aerosol mass both during the bloom (53±16%) and bloom-decay (44±12%) periods. It should be noted that water-soluble OM (WSOM) dominated the mass fraction of OM, where WSOM accounted for 73±15% and 58±22% of OM by mass during the bloom and bloom-decay periods, respectively. Overall, the results showed that the OM was highly water soluble during the study period. For submicrometer aerosols, OC concentration reached up to ~3,000 ngC m<sup>-3</sup>, with averages of 1218±778 ngC m<sup>-3</sup> and 1128±769 ngC m<sup>-3</sup> during the bloom and bloom-decay periods, respectively (**Fig. 3b** and **Table 1**). Average concentrations of WSOC were 777±534 ngC m<sup>-3</sup> (bloom period) and 447±208 ngC m<sup>-3</sup> (bloom-decay period). The concentration levels of WSOC were similar to that (717±440 ngC m<sup>-3</sup>) in the adjacent Oyashio region during the prebloom period (Miyazaki et al., 2020).

Sulfate ( $SO_4^{2-}$ ) was the second most abundant component ( $26\pm10\%$  and  $25\pm4\%$  during the bloom and bloom-decay periods, respectively), followed by ammonium ( $NH_4^+$ ), which accounted for 12% of the submicrometer aerosol mass during both of two periods. The average concentration of sulfate was  $1122\pm744$  ng m<sup>-3</sup> during the bloom and  $1009\pm718$  ng m<sup>-3</sup> during the bloom-decay periods. These concentration levels are approximately eight times higher than that of sulfate ( $130\pm21$  ng m<sup>-3</sup>) in the Arctic region in summer (Ghahreman et al., 2016) and three times (313 ng

m<sup>-3</sup>) higher in high biological activity regions of the Atlantic (O'Dowd et al., 2004). The sulfate concentrations showed temporal and spatial variations similar to those of OC ( $R^2 = 0.60$ ; p < 0.05) and WSOC ( $R^2 = 0.45$ ; p < 0.05) (**Fig. 3c**).

The isotopic composition of aerosol carbon has been successfully used to distinguish the contributions of marine and terrestrial sources to the marine atmosphere (e.g., Cachier et al., 1986). **Fig. 3d** shows temporal changes in the δ<sup>13</sup>C<sub>OC</sub> values during the sailing. Previous studies showed that δ<sup>13</sup>C<sub>OC</sub> values in aerosols of marine origin typically range from -22 to -18‰ (e.g., Crocker et al., 2020). In contrast, typical δ<sup>13</sup>C<sub>OC</sub> values of terrestrial origin are lower than -22‰ (Ehleringer et al., 1997). Among all the collected samples, eleven samples during the bloom period and seven (all) samples during the bloom-decay period showed δ<sup>13</sup>C<sub>OC</sub> values higher than -22‰, indicating a larger contribution of marine origin.

To estimate the relative contributions of marine and terrestrial sources to the OC aerosols, the following massbalance equation was used (Turekian, 2003):

$$\delta^{13}C_{OC} = F_{marine} \times \delta^{13}C_{marine} + F_{terrestrial} \times \delta^{13}C_{terrestrial}$$

where  $F_{\text{marine}}$  and  $F_{\text{terrestrial}}$  are the fractions of contributions from marine and terrestrial sources, respectively, for each sample (i.e.,  $F_{\text{marine}} + F_{\text{terrestrial}} = 1$ ).  $\delta^{13}C_{\text{marine}}$  and  $\delta^{13}C_{\text{terrestrial}}$  are the reported  $\delta^{13}C$  values for marine and terrestrial carbon, respectively. In this study,  $\delta^{13}C_{\text{marine}} = -18\%$  and  $\delta^{13}C_{\text{terrestrial}} = -28\%$  are assumed (e.g., Cachier et al., 1986; Lalonde et al., 2014). Here, if the observed  $\delta^{13}C_{\text{OC}}$  value was higher than -18%,  $F_{\text{marine}}$  was regarded as 100%. **Table 2** shows the calculated fractions of the contributions of marine and terrestrial sources to OC during the bloom and bloom-decay periods. The calculation showed that on average  $73\pm24\%$  and  $82\pm18\%$  of the observed aerosols during the bloom and bloom-decay periods, respectively, were of marine origin (**Table 2**). Indeed, the samples of marine origin showed larger concentrations of OC corresponding to elevated concentrations of Chl *a* particularly during the bloom period (e.g., April 16–18; **Fig. 3a**).

Fig. 4 shows the average mass fraction of each chemical component in submicrometer aerosols of marine origin during the bloom and bloom-decay periods. For aerosols of marine origin, OM accounted for  $48.5\pm13.2\%$  and  $44.0\pm11.5\%$  of the submicrometer aerosol mass during the bloom and bloom-decay periods, respectively, where the difference is insignificant (p = 0.23). In particular, WSOM was dominant OM (75.8±15.0%; p < 0.05) in submicrometer aerosol during the bloom period; the aerosol data of marine origin (i.e.,  $\delta^{13}C_{OC} > -22\%$ ) are used and discussed in terms of WSOC formation in the following sections.

To support the source apportionment, Fig. S2 shows typical 48-hour backward trajectory frequencies calculated from the sampling points for the bloom and bloom-decay periods. The backward trajectory frequencies showed that air

masses with frequencies >40% were indeed transported or originated over the southern Sea of Okhotsk, with minor contributions of land surface (e.g., <20%) such as Hokkaido and eastern Eurasian continent. The trajectory supports the results of  $\delta^{13}$ C values in this study.

# 3.2 Formation process of organic aerosols originated from the sea surface

To investigate the specific origins and formation processes of the observed marine OAs, the relationships between WSOC and several tracers were investigated. Primary aerosols are emitted directly as sea spray aerosols (SSAs) from the ocean surface by wind-driven bubble bursting (Arrigo, 2014). Sodium (Na<sup>+</sup>) is typically used as a tracer for primary SSAs. The Na<sup>+</sup> concentrations and surface wind speeds were positively correlated during the bloom period ( $R^2 = 0.52$ ; p < 0.05) and bloom-decay period ( $R^2 = 0.64$ ; p < 0.05), supporting the hypothesis that Na<sup>+</sup> is a suitable tracer for SSAs.

**Fig. 5** summarizes the coefficients of determination ( $R^2$ ) for WSOC versus the primary and secondary tracers of origin. Time series of the concentrations of each tracer are shown in **Fig. S3**. The WSOC concentrations did not show positive correlations with Na<sup>+</sup> concentrations ( $R^2 = 0.05$  with p = 0.54 during the bloom period and  $R^2 = 0.30$  (negative correlation) with p = 0.12 during the bloom-decay period). These results suggest that the contribution of primary emissions to WSOC mass was not significant during the sailing. Marine secondary aerosols are formed by the atmospheric oxidation of biogenic volatile organic compounds (VOCs), which include phytoplankton-induced dimethylsulfide (DMS), α-pinene, and isoprene (O'Dowd and de Leeuw, 2007; Hallquist et al., 2009). MSA has been widely used as a tracer for marine SOA because it is an oxidation product of DMS. MSA is either produced by gasphase MSA directly scavenged by aerosols or rapidly produced from dimethyl sulfoxide (DMSO) and methanesulfonic acid (MSIA) in the aqueous phase (Bardouki et al., 2003; Lv et al., 2019). The WSOC concentration showed positive correlations with those of MSA (**Fig.S4a**;  $R^2 = 0.62$  and 0.73 (p < 0.05) during the bloom and bloom-decay periods, respectively), suggesting that WSOC, which dominated the OC mass, was affected by the secondary production through the oxidation of DMS or DMS-relevant precursors.

Among other secondary molecular tracers of marine origin, 3-methyl-1,2,3-butanetricarboxylic acid (3-MBTCA) has been recognized as a highly oxidized compound of  $\alpha$ -pinene (Szmigielski et al., 2007). Conversely, 2-methyltetrols (2-MTLs) are oxidation products of isoprene (Claeys et al., 2004). The R<sup>2</sup> values between WSOC and 3-MBTCA concentrations are as high as 0.80 (p < 0.05) and 0.84 during the bloom and bloom-decay periods, respectively (**Fig.S4b**), whereas WSOC and 2-MTLs concentrations did not show any correlation (R<sup>2</sup> < 0.1). This can be explained by the fact that oceanic emission of  $\alpha$ -pinene is typically more evident at high-latitude oceanic regions, whereas the strength of isoprene emission is generally larger at lower latitudes (Zhang et al., 2025). The overall results suggest that

the observed ocean-derived WSOC was affected by secondary formation from DMS-relevant compounds and  $\alpha$ -pinene rather than primary sea spray emissions.

The average MSA and 3-MBTCA concentrations during the bloom-decay period were similar to those during the bloom period (**Table 3**), where the differences are insignificant (p = 0.16 and 0.24 for MSA and 3-MBTCA, respectively). With increasing sunlight intensity, the potential for photochemical reactions in the atmosphere increases, leading to the formation of highly oxidized  $\alpha$ -pinene and DMS-relevant compounds during the bloom-decay period compared to the bloom period (Behrenfeld et al., 2019). Indeed, the average direct solar radiation measured at Abashiri, a coastal city in the southern edge of the Sea of Okhotsk, showed that the intensity during the bloom-decay period (24.7±14.4 MJ m<sup>-2</sup>) was indeed higher (p < 0.05) than that during the bloom period (11.1±7.84 MJ m<sup>-2</sup>) (Japan Meteorological Agency). More dimethylsulfoniopropionate (DMSP) can be released after the bloom climax in late spring, typically resulting in the highest DMS concentrations in the seawater and the atmosphere (Behrenfeld et al., 2019). Together with the correlations shown above, these results suggest that secondary formation, as traced by MSA and 3-MBTCA, is likely responsible for the WSOC concentration levels, even during the bloom-decay period in this study. It should be noted that these tracer compounds accounted for only a small percentage of WSOC or less by mass and that the average concentration of WSOC during the bloom period was larger than that during the bloom-decay period. Therefore, relationships with these tracers only suggest that secondary formations of marine origin were dominant in the observed WSOC aerosols.

#### 3.3 C:N ratios in water-soluble aerosols and surface seawater

A comparison of the C:N ratios between marine-derived aerosols and dissolved organic matter (DOM) in SSW during bloom and bloom-decay periods provides insights into the specific origin and formation processes of marine-derived OAs, which are suggested to be mostly of secondary origin in this study. Throughout the study period, temporal variations in the mass concentrations of WSOC were similar to those of WSON (**Fig. 6**). The WSOC and WSON concentrations showed significant positive correlations during the bloom (R<sup>2</sup>=0.63; p < 0.05) and bloom-decay (R<sup>2</sup>=0.97; p < 0.05) periods. This relationship suggests that these two water-soluble aerosol parameters are strongly associated with a common source on the ocean surface. Moreover, the average concentration of WSON of marine origin during the bloom period (346±240 ngN m<sup>-3</sup>) was more than twice as large as that during the bloom-decay period (130±189 ngN m<sup>-3</sup>). The observed concentration levels were substantially larger than that in total suspended particles observed in the western North Pacific in summer (~20 ngN m<sup>-3</sup>) (Miyazaki et al., 2011). It is noteworthy that C:N ratios in WSOM (WSOC:WSON ratios) during the bloom period (2.6±1.5) were much lower than those during the bloom-decay period (8.5±6.8) (p < 0.05). Furthermore, the WSON concentration (WSOC:WSON ratio) was higher

(lower) (p < 0.05) than those during the pre-bloom period in the Oyashio region (Miyazaki et al., 2018; 2020), a nearby oceanic region of the southern Sea of Okhotsk (**Table S1**). Additionally, EEM spectra in the water-soluble aerosol samples showed that the intensity of the peak at Ex/Em = 275 nm/330 nm, which generally corresponds to protein-like fluorophores, during the bloom (0.024±0.012) was ~1.5 times higher than that in the bloom-decay period (0.015±0.003). These fluorophores are related to marine biological proteinaceous components, including cell fragments, exopolymeric substances, water-soluble amino acids, peptides, and proteins. The lower C:N ratio, together with the larger concentration of WSON and fluorescence intensity of the protein-like compounds, indicated a relatively large contribution of nitrogen-containing OM to submicrometer aerosols of marine origin during the bloom period.

The mass size distributions of WSOC and WSON peaked in the submicrometer size range during both the bloom and bloom-decay periods (**Fig. 7**). The difference in the C:N ratios during the bloom and bloom-decay periods was more significant in the smaller size ranges ( $D_p < 0.58 \mu m$ ; p < 0.05). The peaks measured in the submicrometer size range suggest gas-to-particle conversion of the majority of WSOC and WSON and/or accommodation of VOCs and nitrogenous compounds into the existing particles (e.g., in aqueous phase) rather than being emitted as primary aerosols (i.e., sea spray aerosols). In this study, the  $R^2$  value between WSON and  $Na^+$  concentrations in submicrometer aerosols was below 0.01 (p = 0.05) during the bloom, as expected from the relationship between WSOC and  $Na^+$  concentrations. This also suggests the minor contribution of primary emission of sea spray to WSON, and the major contribution of SOA to WSON in the current study. These results support the secondary formation of WSOC suggested by its relationship with molecular tracers, as discussed previously in Section 3.2.

Fig. 8 shows the average DOC and DON concentrations in the SSW during the bloom and bloom-decay periods.

The average concentrations of DOC (80.3±6.4 μM) and DON (6.5±1.4 μM) during the bloom period were both larger than those during the bloom-decay period (DOC: 65.9±8.6 μM, DON: 4.6±2.2 μM). DON was the dominant component of total dissolved nitrogen (TDN) in SSW, accounting for 80±11% and 60±34% of TDN on a molar mass basis in both the bloom and bloom-decay periods respectively. In SSW, the average C:N ratio of DOM (i.e., DOC:DON ratio) during the bloom period (12.8±2.4) was lower than that during the bloom-decay period (17.3±9.2).

The lower C:N ratio during the bloom period was also found for WSOC and WSON in aerosols as discussed above (Fig. 6), whereas the difference in SSW was not as significant as that found for aerosols. The C:N ratio of the DOM during the bloom period in this study was also lower than that reported during the bloom period in the adjacent Oyashio region (15.9) (Hasegawa et al., 2010). The difference in the C:N ratio between the bloom and bloom-decay periods and between the different oceanic regions may be due to the different characteristics of the centric diatom species, which will be discussed later in this section.

Fig. 9 shows the relative concentrations of WSOC and WSON in aerosols compared to those of DOC and DON in SSW. The linear regression slope of WSOC:WSON in the aerosols during the bloom period (2.13) was lower than that

during the bloom-decay period (2.86), as indicated by the C:N ratio. These slopes of the regression lines were lower than those of DOC:DON in the SSW. Overall, the results suggest that the preferential formation of nitrogen-containing water-soluble OAs (WSOAs) is associated with increased concentrations of DON relative to DOC in the SSW during the bloom period.

# 3.4 Possible factors of marine microbial activity controlling the C:N ratios in water-soluble organic aerosols

The relatively high concentrations of WSON in aerosols and DON in SSW during the bloom period were attributable to marine microbial activity during the bloom in this oceanic region. The most plausible factors controlling these high concentrations were marine diatoms and their exudates. By using seawater samples of 5-m depth during the same sailing, diatoms were dominated by *Thalassiosira hyalina* (0–33%), *Thalassiosira nordenskioeldii* (0–16%), *Fragilariopsis cylindrus* (2–31%), and *Chaetoceros socialis complex* (5–54%) during the phytoplankton bloom of the entire study period (Watanabe, 2022). The diatom genera *Chaetoceros* and *Thalassiosira* dominate marginal ice zone blooms in the subarctic sea (Balzano et al., 2017; Luostarinen et al., 2020). Matsumoto et al. (2021) reported that *Thalassiosira* spp. were dominant (65–95%) from March to the beginning of April, whereas *Chaetoceros* spp. were the most dominant (29–98%) from the mid-April, based on seasonal changes in diatoms in the southern Sea of Okhotsk in 2017. Suzuki et al. (2019) showed that *Thalassiosira nordenskioeldii* and *Chaetoceros* spp. were the dominant diatoms in the Oyashio region of the western subarctic Pacific during spring diatom blooms. Nosaka et al. (2017) suggested an increase in DOC and transparent exopolymer particles (TEPs) by *Thalassiosira nordenskioeldii* in the Oyashio region.

In this study, *Fragilariopsis cylindrus* was observed during the bloom period, whose biomass accounted for a maximum of ~20% of the total biomass (Watanabe 2022). *Fragilariopsis cylindrus* is widespread in polar and subpolar environments and is abundant in both sea ice and the water column (Roberts et al., 2006; Bayer-Giraldi et al., 2010). *Fragilariopsis cylindrus* produces ice-binding proteins (IBPs) and other extracellular polymeric substances (EPS), as well as other DON solutes, including glycine betaine, homarine, and proline (Krell et al., 2007; Boroujerdi et al., 2012; Eickhoff et al., 2023). Polar marine diatoms produce nitrogen-containing compatible solutes in response to salt stress and light stress and can serve as osmoprotectants (Boroujerdi et al., 2012). Although the photosynthetic activity of ice algae increases with increasing light intensity, cell substrates stored in the cell may produce reactive oxygen species that can damage the cell. Consequently, the cells expel excess OM to avoid damage (Kennedy et al., 2021). Indeed, Yan et al. (2020) showed that ice algae release viscous materials from their cells after sea ice melts. Although nitrogenous nutrients were supplied as freshly released DON, this DON may have been affected by rapid bacterial consumption as well as by more intense vertical mixing compared to that during the bloom-decay period. These factors

can explain why the difference in DOC:DON ratios in SSW between the bloom and bloom-decay periods was small compared with those found in aerosols.

Some volatile N-containing compounds associated with marine biota or released from sea ice can be easily transferred from the sea surface to the atmosphere before or without being affected by bacterial consumption or ocean mixing. This process can partly explain the substantially low C:N ratios in the aerosols during bloom periods. Atmospheric emissions of low-molecular-weight alkyl amines associated with sea-ice microbiota have been reported in the Antarctic region (Dall'Osto et al., 2017; 2019). Moreover, Dall'Osto et al. (2017) suggested that the microbiota of sea ice and sea ice-influenced oceans dominated by Fragilariopsis genera can be a significant source of atmospheric ON. However, they did not clarify the relative importance of primary emissions (such as sea spray) or secondary formation of WSON. In summary, the results of this study suggest the preferential formation of nitrogencontaining WSOAs via secondary formation processes associated with the enhanced production of DOC and DON in the SSW of the Sea of Okhotsk during phytoplankton blooms. The enhanced production of DON in the SSW site was likely attributable to the above-mentioned ice algal species during the bloom period. Among the compound groups of WSON, amines are known to trigger new particle formation (Dawson et al., 2012; Almeida et al., 2013), allowing further growth into cloud-seeding particles and affecting the neutralization of aerosol particles. It is noted that the majority of DOC and DON discussed in this study are generally high molecular weight compounds and have low volatility. Therefore, photodegradation and/or biodegradation of DOC and DON in the air-sea interface are likely important to produce more volatile compounds for the atmospheric emissions, which needs further investigation in 410 future studies. The results of this study provide insights into the importance of the sea ice region as a source of OAs, which has not yet been sufficiently considered.

## **4 Conclusions**

We investigated how oceanic biological activity during phytoplankton blooms in the Sea of Okhotsk affects the formation of atmospheric OAs based on ambient submicrometer aerosols and surface seawater samples collected in the oceanic region during springtime blooms and bloom-decay periods. Chemical analysis of aerosol samples revealed that OM was the dominant component of submicrometer aerosols during both the bloom (53±16%) and bloom-decay periods (44±12%). In particular, OM was found to be more water soluble during the bloom period, when WSOM accounted for 73±15% of OM by mass.

Analysis of  $\delta^{13}C_{OC}$  showed that 73±24% and 82±18% of the observed aerosols were of marine origin (> -22%). For aerosols of marine origin, OM accounted for 48% and 45% of the submicrometer aerosol mass during the bloom and bloom-decay periods, respectively, whereas the average OC concentrations did not show significant differences between the two periods. Correlations between concentrations of WSOC and those of molecular tracers suggested that the ocean-derived WSOC observed in this study was likely affected by secondary formation rather than primary emissions of sea spray aerosols during the study period.

The average C:N ratio in WSOAs of marine origin was significantly lower during the bloom period  $(2.6\pm1.5)$  than during the bloom-decay period  $(8.5\pm6.8; p < 0.05)$ . These results suggest the preferential formation of nitrogen-containing WSOAs in aerosols associated with marine microbial activity during blooms. Indeed, the average concentrations of DOC and DON in surface seawater during the bloom period were higher than those during the bloom-decay period. The relatively large amount of DON followed by the preferential formation of N-containing WSOM aerosols during the bloom period was likely associated with the predominant diatoms *Thalassiosira* spp. and *Fragilariopsis* spp. in the southern Sea of Okhotsk during the study period.

This study highlights the significant contribution of secondary formation of marine biogenic nitrogen-containing OAs during the phytoplankton bloom after sea ice retreat in the Sea of Okhotsk. The secondary formation of marine organic aerosols linked with ice algae has not been extensively considered in climate models, and the current study highlights the importance of this atmospheric formation process of OM originating from the sea surface after ice melting in the subarctic ocean.

# Data availability

The measurement data for the aerosol and seawater samples are provided in the Supplement. All the other data are available upon request.

#### **Author contributions**

YM designed the study, and YW and YM wrote the manuscript. YW, KS, YY, and Y.M. collected samples. YW, ET, YM, YY, KS, and JN performed the measurements and analyses of the aerosol and seawater parameters. JN designed and managed all measurements performed during the field observations.

## 445 Competing interests

The authors declare that they have no conflict of interest.

#### Acknowledgements

We thank all the crews and colleagues for helping with sampling during the R/V *Shinsei Maru* KS-21-6 (JURCAOSS21-15) sailing. We also thank A. Kamimura and A. Murayama for their help in the measurements of biological and chemical parameters in seawater.

## **Financial supports**

This research was supported by Grants-in-Aid for Scientific Research (19H04233, 21H05056, 23K28206, 23K28207) from the Ministry of Education, Culture, Sports, Science, and Technology (MEXT), Japan, Grant-in-Aid for Transformative Research Areas (22H05205), and Grant for Joint Research Program of the Institute of Low Temperature Science, Hokkaido University.

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

**Table 1.** Average concentration of each parameter in aerosol of marine origin and surface seawater during the bloom and bloom-decay periods. Shown are average values and standard deviations. Numbers in the parenthesis are for all the data including terrestrial origin.

|                                    | Bloom period           | Bloom-decay period |
|------------------------------------|------------------------|--------------------|
| OC (ngC m <sup>-3</sup> )          | 1194±682<br>(1218±778) | 1128±769           |
| WSOC (ngC m <sup>-3</sup> )        | 803±555<br>(777±534)   | 447±208            |
| WIOC (ngC m <sup>-3</sup> )        | 391±243<br>(441±349)   | 681±605            |
| WSON (ngN m <sup>-3</sup> )        | 346±240<br>(283±231)   | 130±189            |
| WSOC:WSON                          | 2.6±1.5<br>(3.2±1.9)   | 8.5±6.8            |
| Sulfate (ng m <sup>-3</sup> )      | 1228±690<br>(1100±671) | 1009±718           |
| Sodium (ng m <sup>-3</sup> )       | 42±30<br>(50±35)       | 142±100            |
| DOC:DON                            | 12.8±2.4               | 17.3±9.2           |
| Chl <i>a</i> (mg m <sup>-3</sup> ) | 5.58±2.05              | 4.00±1.77          |

**Table 2.** Fractions of contributions of marine  $(F_{marine})$  and terrestrial  $(F_{terrestrial})$  sources to the observed OC in submicrometer aerosols during the bloom and bloom-decay periods, which were calculated by stable carbon isotope ratios of total carbon (TC).

|                         | Bloom period | Bloom-decay period |
|-------------------------|--------------|--------------------|
| $F_{marine}$            | 74±25%       | 92±32%             |
| F <sub>terretrial</sub> | 26±25%       | 8±32%              |

**Table 3.** Average concentrations of molecular tracers of biogenic secondary aerosols of marine origin during the bloom and bloom-decay periods. Shown are average values and standard deviations. Numbers in the parenthesis are for all the data including terrestrial origin.

| Tracers (ng m <sup>-3</sup> )                            | Bloom period             | Bloom-decay period |
|----------------------------------------------------------|--------------------------|--------------------|
| 2-methyltetrol                                           | 0.08±0.06<br>(0.07±0.06) | 0.11±0.09          |
| 3-methyl-1,2,3-<br>butanetricarboxylic<br>acid (3-MBTCA) | 0.96±0.73<br>(0.87±0.93) | 1.27±1.90          |
| MSA                                                      | 56±44<br>(50±42)         | 79±47              |

Figure 1: (a) Monthly average chlorophyll *a* (Chl *a*) concentrations in April 2021, derived from the Aqua-MODIS (https://oceandata.sci.gsfc.nasa.gov/l3/), with (b)(c) the track of the R/V *Shinsei Maru* KS-21-6 sailing in the southern Sea of Okhotsk from April 11 to May 1, 2021. The red color in (b) and (c) represents the track during the first half of the sampling period (April 13–20, 2021), which was defined as the bloom period in this study. The blue color represents the track during the second half of the sampling period (April 24–27, 2021), which was defined as the bloom-decay period. The black line indicates the track outside the bloom and bloom-decay periods.

Figure 2: Time series of Chl a concentration in surface seawater during the expedition. The red and blue arrows represent the bloom and bloom-decay periods defined in this study.

Figure 3: Timeseries of (a) the mass concentration and fraction of each chemical species, (b) OC and WSOC concentrations, (c) sulfate concentrations, and (d) the  $\delta^{13}$ C value of OC ( $\delta^{13}$ C<sub>OC</sub>) in submicrometer aerosols observed in the Sea of Okhotsk during the KS-21-6 expedition.

Figure 4: The average mass fraction of each chemical component in the submicrometer aerosols of marine origin during the bloom (left) and bloom-decay periods (right).

Figure 5: R<sup>2</sup> values between concentrations of primary/biogenic secondary tracers and WSOC in the submicrometer aerosols for the bloom and bloom-decay periods. Note that the concentration of WSOC showed negative correlations with that of Na<sup>+</sup>, which is indicated as (–).

Figure 6: Timeseries of the mass concentration of WSOC and WSON in submicrometer aerosols. Dotted lines indicate the data with  $\delta^{13}C_{WSOC} 

Figure 7: Average size distributions of the (a) WSOC and (b) WSON mass, and the (c) WSOC/WSON mass ratio during the bloom (red) and bloom-decay (blue) periods.

Figure 8: Average concentrations of (a) dissolved organic carbon (DOC) and (b) dissolved organic nitrogen (DON) in surface seawater during the bloom and bloom-decay periods. Bars indicate standard deviations.

Figure 9: Scatter plots of WSOC vs. WSON in submicrometer aerosols (solid circles) and DOC vs. DON in surface seawater (open circles). Data obtained during the bloom period are shown in red and those during the bloom-decay period are shown in blue.