# Peer review of "Formation of marine atmospheric organic aerosols associated with the spring phytoplankton bloom after sea ice retreat in the Sea of Okhotsk"

_EGUsphere, 2025_

## Author Comment (AC1)

**Responses to the comments of Referee#1**

**General comments:**

The organic component of aerosol is an important and uncertain aspect of aerosol composition, particularly in the marine atmosphere. Given the importance of aerosols in atmospheric chemistry and climate developing a better understand of this aerosol organic matter is valuable and this paper is a useful contribution to this goal.

This is a thorough and interesting study of aerosols during the spring bloom period in the Sea of Okhotsk. The chemical characterisation particularly of the aerosol is comprehensive, sophisticated. and well described. Overall I believe the paper is well worth publication but I do have some suggestions for modifications prior to final publication.

Reply: We appreciate the referee's valuable comments on our work. Our responses to the specific comments and details of the changes made to the manuscript are given below.

**Specific comments:**

Firstly I believe it would be useful to include some further descriptions of the conditions at the time of sampling.

1. There is talk of ice algae and I'm not clear whether thee was a lot of ice at the time of sampling or not. This is potentially important because an ice cap can allow a build up of quite high concentrations of marine biogenic gases which are then released rapidly as the ice breaks up.

Reply 1: During the sampling period of this study, sea ice had already retreated, and there was no ice around the observed region. To show the special distributions of sea ice and their temporal changes, the maps of sea ice extent in the Sea of Okhotsk in March and April 2021 are now added as Fig. S1 in the Supplement. Also, the corresponding sentence has been revised as follows:

L.68: "During this period when the sea ice had already melted or retreated (Fig. S1), .."

2. What were the wind conditions like? – this is relevant to ice break up, seawater mixing and bloom development and to seaspray emissions.

Reply 2: The average local wind speeds during each sampling duration were ~4–12 m s-1 as shown below. Because sea ice had already retreated during the study period, as mentioned above, the local wind speeds were unlikely to be relevant to ice breakup during the study period. Instead, the wind speed was relevant to the atmospheric emission of sea spray aerosols, as the referee pointed out. The figure below illustrates that Na+ concentrations in submicrometer aerosol and local wind speeds showed significant positive correlations both during the bloom ( $R^2 = 0.52$ ; p < 0.05) and bloomdecay ( $R^2 = 0.64$ ; p < 0.05) periods. This supports that  $Na^+$  concentrations in submicrometer aerosols can be a suitable tracer of sea spray in this study. This information is already described in the original manuscript, but the sentence has been modified in the revised manuscript (L. 267: "The Na+ concentrations and surface wind speeds were positively correlated during the bloom period ( $R^2 = 0.52$ ; p

Figure: Scatterplot between sodium concentration in submicrometer aerosols vs. local wind speeds. The values of local wind speed are averages during each aerosol sampling duration and bars indicate standard deviations. The data obtained during the bloom and bloom-decay periods are shown in red and blue, respectively. Solid circles indicate the data of marine origin defined by stable carbon isotope ratios in submicrometer aerosols.

3. The apparently very low contribution of terrestrial derived atmospheric aerosol organic matter leads to a question of where the air came from during the sampling period?, so including some air-parcel back trajectories would be useful.

Reply 3: As the referee suggested, we have added representative backward trajectories calculated from the sampling points during the bloom and bloom-decay periods (as Figure S2) to show the origins of the typical air mass. The backward trajectory frequencies showed that air masses with frequencies >40% were indeed transported or

originated over the southern Sea of Okhotsk, with minor contributions (e.g.,

Figure S2: Typical 48-hour backward trajectory frequencies calculated from the sampling points for the bloom and bloom-decay periods.

The corresponding statement on the trajectory has been added to the text in the revised manuscript.

L.257: "...To support the source apportionment, Fig. S2 shows typical 48-hour backward trajectory frequencies calculated from the sampling points for the bloom and bloom-decay periods. The backward trajectory frequencies showed that air masses with frequencies >40% were indeed transported or originated over the southern Sea of Okhotsk, with minor contributions (e.g., <20%) from land surfaces, such as Hokkaido and eastern Eurasian continent. The trajectory supports the results of  $\delta^{13}C$  values in this study."

Throughout the discussion the authors should be clear which size of aerosol particles they are discussing. I became confused at several points. Gas phase emissions from seawater will form fine mode particles, while ejection of seawater itself will produce coarse mode particles. Some of the correlations such as in Figure 5 are not really useful given these differences.

Reply 4: In this study, we focused on the submicrometer particle as its size range is important for CCN activity. Supermicrometer particles or coarse mode particles are not discussed in this study, because their atmospheric residence time is much shorter than that of submicrometer particles. Moreover, the size distributions of WSOC shown in Figure 7 clearly support the importance of submicrometer particles in terms of the formation process of WSOC in this study. Based on these, we believe that Figure 5 is

still useful to discuss the origin and formation processes of submicrometer particles that can contribute to CCN.

In the revised manuscript, we have clearly mentioned that (L. 76) "we focused on the analytical results obtained from the bottom stage of the impactor, which collected particles with an aerodynamic diameter ( $D_p$ ) of < 0.95  $\mu$ m and are referred to as submicrometer particles."

Additionally, we added the following statement at the end of section 2.2: (L. 92) "In this study, the results of submicrometer particles collected by the HVAS are mainly shown, whereas the results of aerosol particles collected by the cascade impactor are presented only to show the size distributions of water-soluble fractions of organic matter (section 3.3)."

Also in the captions of Figures 5, 6, and 9, the words "submicrometer aerosols" have been added.

Section 3.2 is a bit misleading. As the authors correctly note at the end of this section (line 271-2) the tracer species they use represent only a tiny fraction of the WSOM and so the origin of this material is still essentially unknown, although the correlations to MSA and 3MBTCA are intriguing. I would suggest reorganising this section to avoid any misunderstandings over what can and cannot be said about the sources of the WSOM.

Reply 5: As the referee pointed out, we cannot identify the chemical structure or compositions of the majority of WSOM here, only from the relations of WSOM with those molecular tracers. Nevertheless, the correlations with 3-MBTCA and MSA together with a lower correlation with sodium at least suggest secondary formation of OA of marine biogenic origin rather than primary sea spray as the formation process of WSOC. That is what we intend to emphasize. In the revised manuscript, some sentences that were overstated have been modified as follows:

L.279: "The WSOC concentration showed positive correlations with those of MSA (Fig.S4a;  $R^2 = 0.62$  and 0.73 (p < 0.05) during the bloom and bloom-decay periods, respectively), suggesting that WSOC, which dominated the OC mass, was affected by the secondary production through the oxidation of DMS or DMS-relevant precursors."

L.289: "The overall results suggest that the observed ocean-derived WSOC was affected by secondary formation from DMS-relevant compounds and  $\alpha$ -pinene rather than primary sea spray emissions."

I was also a little confused by the logic of the argument in sections 3.3 and 3.4. The DOC and DON in seawater is overwhelmingly of high molecular weight and long lived. The observed relationships of DOC and DON in seawater (Fig 9) reflect the fact that they are probably actually bonded together in the same complex organic matter and the variations in concentrations in both compounds may reflect changes in production and consumption, or alternatively may reflect physical mixing of water masses. The correlations of DOC and DON in the aerosols look less convincing in Figure 9, and this correlation too could also represent mixing of air masses. Given its molecular weight, the direct emissions of seawater DOC and DON into the atmosphere will be via bubble bursting type processes and hence associated with coarse mode aerosol, as with sodium. This process cannot therefore explain the fine mode WSOM or the relationships of WSOM to MSA and other gaseous marine biogenic emissions reported here. All the data I have seen published suggests that marine amine emissions are very small, particularly in comparison to say ammonia emissions. Hence the emission of gaseous organic compounds from seawater into the atmosphere does not seem to be able to explain aerosol DON, although it could arise from marine biogenic gas emissions of other non-nitrogenous compounds with nitrogen being subsequently incorporated during aerosol formation. So I find the authors observations valuable and interesting, I am not sure they do provide a clear explanation of the formation mechanism for the aerosol WSON as implied particularly in the abstract. I would suggest that the logic of the argument in sections 3.3 and 3.4 might therefore be clarified.

Reply 6: If the direct atmospheric emissions of seawater DOC and DON via bubble bursting processes were significant to form the observed WSOC and WSON, their size distributions should show a dominant mode in the supermicrometer size range (or coarse mode). However, as shown in Figure 7, the mass of WSOC and WSON of marine origin resided mostly in the submicrometer size range in this study. This result of size distribution, together with the correlation with molecular tracers suggested the secondary formation of WSON rather than primary emissions with the dominant mode in supermicrometer size or coarse mode particles.

We agree that the sea-to-air emissions of amines are small compared to that of the bulk WSON and ammonia. In the current discussion, amines are raised as a candidate compound group associated with sea-ice microbiota, but they are not regarded as a major compound group of WSON. As the referee pointed out, it is possible that marine biogenic gas emissions of other non-nitrogenous VOCs, along with ammonia or reactive nitrogen, are subsequently converted to particles and/or incorporated into the existing particles (e.g., aqueous phase). As it is difficult to provide a clear explanation of the exact mechanism for the aerosol WSON formation in this study, we can just describe it as (L.354) "the preferential formation of N-containing water-soluble OAs ... during the bloom period."

Taking into account the comment, we revised the statement in section 3.3 regarding the points above as follows:

L. 331: "... The peaks measured in the submicrometer size range suggest gas-to-particle conversion of the majority of WSOC and WSON and/or accommodation of VOCs and nitrogenous compounds into the existing particles (e.g., in aqueous phase) rather than being emitted as primary aerosols (i.e., sea spray aerosols). In this study, the  $R^2$  value between WSON and  $Na^+$  concentrations in submicrometer aerosols was below 0.01 (p = 0.05) during the bloom, as expected from the relationship between WSOC and  $Na^+$  concentrations. This also suggests the minor contribution of primary emission of sea spray to WSON, and the major contribution of SOA to WSON in the current study. These results support the secondary formation of WSOC suggested by its relationship with molecular tracers, as discussed previously in Section 3.2."

---

## Author Comment (AC2)

**Responses to the comments of Referee#2**

Comments: This is a review of the manuscript: "Formation of marine atmospheric aerosols associated with the spring phytoplankton bloom after sea ice retreat in the Sea of Okhotsk" by Miyazaki et al. The manuscript refers to samples collected during a field campaign in April 2021 during the bloom and post-bloom periods. The authors want to investigate the effects of marine phytoplankton emissions on the atmospheric organic aerosol composition. The results show an increase in WSON (and an associated decrease in the C:N ratio) and changes in WSOC, although bloom values are very similar to pre-bloom values from a previous study.

While the field of research is extremely interesting, I am afraid that the experimental set-up is not ideal, specifically because a pre-bloom period should have been included to better understand the effects of phytoplankton on altering the molecular composition of OA. Considering the limited number of days investigated (six) during the bloom-decay period, who can tell if the observed changes in WSON and WSOC are related to phytoplankton activity or simply internal variability (e.g., changes in air-mass sources)?

On top of that, I think that the manuscript suffers from key methodological issues: the authors used sonication to extract their organics. However, it is known that organics can degrade during sonication. Did the authors can prove that degradation of the targeted organic molecules is absent? Also, where the filter punches were located before ultrasonication? In plastic vials? How were they cleaned? This is critical as ultrasonication can release impurities from the vial walls if not properly cleaned. How many blanks were collected? How they looked like (this holds for both inorganic and organic species)? Other missing methodological details include: a) at which temperature and vacuum the rotary evaporator was operating? b) How the method for organics was validated (i.e., what is the recovery and reproducibility of the extractions)?

More general and specific comments are reported below together with some suggestions that the authors can use to improve their manuscript.

Overall, my recommendation is **major revisions** and reconsideration for publication when the concerns are properly addressed.

Reply 1: We appreciate the referee's valuable comments on our work. We provide a point-by-point response to each comment/question raised above as follows.

-While the field of research is extremely interesting, I am afraid that the experimental setup is not ideal, specifically because a pre-bloom period should have been included to better understand the effects of phytoplankton on altering the molecular composition of OA. Considering the limited number of days investigated (six) during the bloom-decay period, who can tell if the observed changes in WSON and WSOC are related to phytoplankton activity or simply internal variability (e.g., changes in air-mass sources)?

Reply 2: We have the aerosol data obtained during the pre-bloom period in the Oyashio region (Miyazaki et al., 2018; 2020), a nearby oceanic region of the southern Sea of Okhotsk. Taking account of the comments, we have provided the major aerosol chemical parameters of "marine origin" observed during the pre-bloom period to compare them with those during the bloom and bloom-decay periods of this study in the revised manuscript (as Table S1). At least, the differences in the WSON concentration, WSOC:WSON ratio, and Chl  $\alpha$  concentrations in surface seawaters between the pre-bloom and bloom periods were significant (p < 0.05), supporting our conclusion in this study. Also, we made an additional statement on this in the revised manuscript.

L.320: "...Furthermore, the WSON concentration (WSOC:WSON ratio) was higher (lower) (p < 0.05) than those during the pre-bloom period in the Oyashio region (Miyazaki et al., 2018; 2020), a nearby oceanic region of the southern Sea of Okhotsk (Table S1)."

- On top of that, I think that the manuscript suffers from key methodological issues: the authors used sonication to extract their organics. However, it is known that organics can degrade during sonication. Did the authors can prove that degradation of the targeted organic molecules is absent?

**Reply 3: Please see our Reply 22 to the duplicated comment below.**

-Also, where the filter punches were located before ultrasonication? In plastic vials? How were they cleaned?

Reply 4: The filter punch was located in glass and plastic vials for the analysis of organic and inorganic species, respectively. All the vials were cleaned with detergent, followed by being washed with distilled water. The glass vials were then baked at 410°C for two hours to remove VOC contaminants prior to the extraction process. The following statement has been added to the text in the revised manuscript:

L.80: "The filter punch was located in glass and plastic vials for the analysis of organic and inorganic species, respectively. All the vials were cleaned with detergent and then washed with distilled water. The glass vials were then baked at 410 °C for two hours to remove contaminants prior to the extraction process."

-How many blanks were collected? How they looked like (this holds for both inorganic and organic species)?

Reply 5: Blank filters were collected four times on the ship during the expedition. The blank value was calculated from their average mass, which accounted for the mass of <30% of WSOC and WSTN, <27% of inorganic ions, and <3% of the organic compounds measured in this study. All the mass concentrations were calculated by subtracting the original values from these blanks. The statement on the blanks is now additionally made in the revised manuscript (L.80, 120, 127, 168).

-a) at which temperature and vacuum the rotary evaporator was operating?

Reply 6: The rotary evaporator was run at 25°C, and the gauge on the vacuum pump read 80 mmHg. This sentence is added to the text in the revised manuscript (L.158).

- b) How the method for organics was validated (i.e., what is the recovery and reproducibility of the extractions)?

Reply 7: The recovery of each organic compound is higher than 81% for all the compounds measured, and the reproducibility of the extraction is <13%. This information is added to the text in the revised manuscript (L.167).

Further responses to the specific comments and details of the changes made to the manuscript are given below.

**General comments:**

• The authors used an HVAS and an Andersen to collect atmospheric aerosol samples. However, it is not clear in the text which filters were analyzed for what.

Reply 8: Basically, the analytical results of the bottom filter samples (submicrometer aerosol samples) obtained by HVAS are shown throughout the manuscript, whereas the results obtained by the Andersen impactor are only shown in Fig.7 (the 1st half of section 3.3) to emphasize the importance of the submicrometer size range in terms of secondary formation of WSOC in this study. This statement has been now added to the text of the revised manuscript as follows:

L.92: "In this study, the results of submicrometer particles collected by the HVAS are mainly shown, whereas the results of aerosol particles collected by the cascade impactor are presented only to show the size distributions of water-soluble fractions of organic matter (section 3.3)."

Comparisons between averages were done mainly qualitatively. However,
differences must be either significant or not significant with associated p-values. I
suggest the authors implement the appropriate statistical tests to discuss differences
between averages throughout the entire manuscript. Also, p-values must be
provided for all the correlation values discussed in the text.

Reply 9: According to the comment, we have just provided p-values for all the relevant values to discuss each difference in the text. Further details are given in each reply below.

• From my understanding at L138-140, the authors say that they analyze oxidation products of a-pinene (3-MBTCA, pinic acid and pinonic acid) and of isoprene (2-methyltetrols). Also glucose was mentioned, but not analyzed. Why do they only discuss 3-MBTCA and 2-methyltetrols?

Reply 10: In this manuscript, a representative compound of each tracer is shown. 3-MBTCA and 2-methyltetrols are shown as representative compounds of oxidation products of  $\alpha$ -pinene and isoprene, respectively. Glucose was not shown here as its characteristics were similar to those of sodium, which serves as a tracer of primary emission from the sea surface. Taking into account the comments, the compounds for which the results are not shown in the manuscript (i.e., pinic acid, pinonic acid, and glucose) have been deleted from the text.

• There is a lot of discussion regarding WSOM, but not much regarding the WIOM. Whether I can understand the reason why WSOM increases during the bloom period (more oxidation of volatile precursors), why WIOM increases during the bloom-decay period (almost doubling)?

Reply 11: As the referee pointed out, the increase in the WIOM concentration was observed during the bloom-decay period, particularly on April 23–25 (Fig. 3a). Considering the large contribution of marine source based on the stable carbon isotope ratios, one possible explanation for the increase is that the contribution of sea spray aerosol to WIOM was likely significant compared to that during the bloom period. This is partly supported by the increase in the Na+ concentration, which ranged from ~60 to 300 ng m-3 during the corresponding period (Fig.S2a), although a positive relationship between WIOM and Na+ concentrations is not clear due to the limited number of samples.

• I don't fully agree with the conclusions: "the current study highlights the importance of this atmospheric formation process of OM originating from the sea surface after ice melting in the subarctic region". How, in the absence of data regarding the pre-bloom period? I see that isotope analyses and correlations with MSA can be supportive of this conclusion, but at the same time we do not have a clear chemical characterization of the conditions before the bloom.

**Reply 12: Please see our Reply 2 to the duplicated comment above.**

**Specific comments:**

L17-18: I would downsize the emphasis of this sentence: "Relations between WSOC and molecular tracers suggested that the majority of WSOC of marine origin was affected [...] instead of primary emissions of sea spray aerosol". Here the authors based their interpretation on a linear correlation between WSOC and two molecules (3-MBTCA and MSA). Considering that WSOC is composed of several hundreds of different compounds, I think that this is a bit over interpretation.

**Reply 13: According to the comment, we have weakened the statement as follows:**

L.17: "Relations between water-soluble OC (WSOC) and molecular tracers suggested that aerosol WSOC of marine origin was likely affected by secondary formation from precursors such as  $\alpha$ -pinene and DMS-relevant compounds rather than primary emissions of sea spray aerosols."

L27–28: I would reformulate the sentence. Oceans are sources of volatile organic compounds, not of "secondary organic aerosols," which indeed form in the atmosphere through reaction with atmospheric oxidants.

**Reply 14: Taking into account the comment, the sentence has been revised as follows:**

L.27: "Ocean surfaces are a significant source of atmospheric organic aerosols (OAs), which are formed by direct emissions of particles (i.e., sea-spray aerosols) or by volatile organic compounds (VOCs) followed by particle formation (i.e., secondary OAs)."

L65: could you also add a sea-ice map for the corresponding year, showing where the sea-ice was and at which concentration (March-April would be enough).

Reply 15: According to the comment, maps showing the temporal change in the seaice extent in the Sea of Okhotsk from March to April 2021 (every 10 days) have been added as Fig. S1 in the Supplement.

L68–69: I would put letters a, b, c in the panels of Figure 1 and refer to Fig. 1a etc. in the main text.

Reply 16: The letters a, b, and c have been inserted in Figure 1, and they are referred to as Fig. 1a, 1b, and 1c in the text as suggested.

L80: Can the authors better clarify which sizes the nine-stage size-segregated aerosol samples refer to?

Reply 17: The size ranges of aerosol particles collected by the Andersen impactor are now shown in section 2.2 of the revised manuscript (L.86).

L70–87: It is not clear to me which aerosol fractions were analyzed for what, as in the results the authors do not refer to any specific aerosol fraction. Also, from which fractions were the organic tracers MSA, 3-MBTCA, 2-MTL, and Na+ analyzed? Based on what the authors write at L101 ("submicrometer sample"), I guess these analyses were done from the HVAS filters, but I would appreciate better clarification.

Reply 18: Throughout the manuscript, we focused on the analytical results obtained from the bottom stage of the impactor, which collected submicrometer particles. The only exception is Figure 7, which shows the mass size distributions of WSOC and WSON across all size ranges to support the discussion on the secondary formation of those parameters. In the revised manuscript, we have clearly mentioned this point as follows:

L.76: "...we focused on the analytical results obtained from the bottom stage of the impactor, which collected particles with an aerodynamic diameter (Dp) of  $

**Fig.** Concentrations of 2-methyltetrols and 3-MBTCA derived from the sample filters under different conditions of extraction: only shaking with no ultrasonication (US), 5 mins, 10 mins, and 20 mins of US.

Ultrasonication is an important procedure for the complete extraction of target compounds from a sample filter. As evidence, extraction with just shaking and ultrasonication for a short time (5 mins in this case) resulted in the lower concentration measured, which suggests incomplete extraction rather than decomposition of compounds. At least the results did not show the effect of decomposition of the target compounds. Based on this experiment, we believe that ultrasonication for 10 mins (as our method) is an appropriate condition to extract the compounds in our study.

L129: Filter cut from HVAS? Andersen? Please specify here and elsewhere.

Reply 23: It is a filter cut from the HVAS. In the revised manuscript, it is now clearly mentioned in that part (L.142) as well as in all the other parts (L.109, 122, 132, 154, 172).

L139–141: As far as I can understand, the authors report that tracer compounds such as 3-MBTCA, pinic acid, and pinonic acid were analyzed. I was wondering why only 3-MBTCA is reported and not pinic acid and pinonic acid. Ratios between pinic acid and 3-MBTCA could also be particularly interesting from an atmospheric chemistry point of view to investigate how atmospheric aging proceeds.

Reply 24: As we explained in Reply 10, a representative compound of each tracer is shown in this manuscript. 3-MBTCA and 2-methyltetrols are shown as representative compounds of oxidation products of  $\alpha$ -pinene and isoprene, respectively. In fact, the temporal variations in the concentrations of 3-MBTCA, pinic acid, and pinonic acid were similar, and 3-MBTCA is shown as a representative compound of oxidation products of  $\alpha$ -pinene. Also, because ratios of pinonic acid (or pinic acid) to 3-MBTCA did not show any statistical difference between the bloom and bloom-decay periods, we did not include pinic acid or pinonic acid in our manuscript. As we mentioned in Reply 10, the compounds for which the results are not shown in the manuscript (i.e., pinic acid, pinonic acid, and glucose) have been deleted from the text.

L151: Which internal standard? At which concentration? Was an internal standard also added before rotary evaporation to assess any potential loss of the analytes or was it only used to monitor instrument performances?

Reply 25: As the internal standard, n-Tridecane ( $C_{13}$ ) with a concentration of 1.43 ng  $\mu l^{-1}$  in hexane was added just before the injection into GC-MS to determine the mass of target compounds, considering the status of the instrument. We have just added this information to the text as follows:

L.163: "After the derivatization, the derivatives were diluted with hexane containing the internal standard (n-Tridecane ( $C_{13}$ ) with concentration of 1.43 ng  $\mu l^{-1}$  in hexane) and then injected into a capillary gas chromatograph (GC8890, Agilent) coupled to a mass spectrometer (MSD5977B, Agilent)..."

L203-204: What can be the interpretation of these sulfate values? Can be a contribution from fossil sources a reason?

Reply 26: Based on the stable carbon isotope ratios together with the positive correlation between WSOC and sulfate, the increase in sulfate concentration was

attributable to marine sources (i.e., DMS) rather than anthropogenic sources such as fossil fuel combustion. Indeed, the concentration of sulfate showed a positive correlation with that of MSA ( $R^2$  =0.61; p < 0.05) during the entire period, which supports this explanation.

L206: what is the correlation coefficient between sulfate, and OC and WSOC? This information should be added to add consistency to the discussion.

Reply 27: As the referee pointed out, the coefficients of determination for sulfate vs. OC ( $R^2 = 0.60$ ; p < 0.05), and sulfate vs. WSOC ( $R^2 = 0.45$ ; p < 0.05) have been added to the text (L.228).

L224–225: While the isotope analyses are undoubtedly interesting, I would integrate also some back-trajectory studies in order to corroborate these findings for both investigated periods (e.g., 72 hrs). This would provide additional evidence to the isotope results in clearly showing the air-mass provenance, and proving that changes in WSON are associated to changes in the marine environment and not to changes in air mass provenance.

Reply 28: As the referee suggested, we added representative backward trajectories for the bloom and bloom-decay periods (as shown in Figure S2) to illustrate the origins of the typical air masses transported in this study. The backward trajectory frequencies showed that air masses with frequencies >40% were indeed transported or originated over the southern Sea of Okhotsk, with minor contributions (e.g.,

Figure S2: Typical 48-hour backward trajectory frequencies calculated from the sampling points for the bloom and bloom-decay periods.

The corresponding statement on the trajectory has been added to the text.

L.257: "To support the source apportionment, Fig. S2 shows typical 48-hour backward trajectory frequencies calculated from the sampling points for the bloom and bloom-decay periods. The backward trajectory frequencies showed that air masses with frequencies >40% were indeed transported or originated over the southern Sea of Okhotsk, with minor contributions (e.g., <20%) from land surfaces, such as Hokkaido and eastern Eurasian continent. The trajectory supports the results of  $\delta^{13}C$  values in this study."

L227–232: I would probably avoid a pie chart considering the great variability of the dataset. Discussing average values without their standard deviation doesn't say much about whether the observed differences are actually significant.

Reply 29: We believe that the pir charts shown in Fig. 4 are still useful for identifying the average fraction of chemical components shown in Fig. 3a, so we have decided to keep them.

For Fig.4, what we intend to mention is:

- 1) WSOM was dominant OM during the bloom period (75.8 $\pm$ 15.0%; p < 0.05).
- 2) The difference in the OM/submicrometer mass ratio during the bloom  $(48.5\pm13.2\%)$  and bloom-decay  $(44.0\pm11.5\%)$  periods was insignificant (p=0.23).

As the referee pointed out, we have added the standard deviations to indicate statistical significance in the corresponding statement as follows:

L.252: "For aerosols of marine origin, OM accounted for  $48.5\pm13.2\%$  and  $44.0\pm11.5\%$  of the submicrometer aerosol mass during the bloom and bloom-decay periods, respectively, where the difference is insignificant (p=0.23). In particular, WSOM was dominant OM ( $75.8\pm15.0\%$ ; p<0.05) in submicrometer aerosol during the bloom period;.."

L243 and L250 and L257: when the authors mention the word "significant" or "insignificant" they should also provide a p-value.

Reply 30: Among the lines, the word "insignificant" was used only in L.243, but not in L.250 and 257 in the original manuscript. Because p-values were 0.54 and 0.12, we modified the sentence as follows:

L. 271: "The WSOC concentrations did not show positive correlations with  $Na^+$  concentrations ( $R^2 = 0.05$  with p = 0.54 during the bloom period and  $R^2 = 0.30$  (negative correlation) with p = 0.12 during the bloom-decay period)."

L249–252: It would be nice to see scatter plots of the correlations between WSOC and MSA and 3-MBTCA as well as the time series. Also, this sentence should be downsized: a simple correlation with WSOC and MSA, while suggesting a process, cannot really be generalized in the way the authors claimed ("greatly affected"). As I wrote elsewhere, organic aerosol is constituted up to several thousands of compounds.

Reply 31: According to the comment, scatterplots of WSOC vs. MSA and WSOC vs. 3-MBTCA have been added to the supplement as Fig. S4. In addition, the sentence has been revised (downsized) as follows:

L. 279: "The WSOC concentration showed positive correlations with those of MSA (Fig. S4a; R2 = 0.62 and 0.73 (p < 0.05) during the bloom and bloom-decay periods, respectively), suggesting that WSOC, which dominated the OC mass, was affected by the secondary production through the oxidation of DMS or DMS-relevant precursors."

L259–L261: I believe that this sentence should be downsized, as it is very hard to say if the lack of correlation with methyltetrols can really be associated with a lack of isoprene emissions from the ocean water. While being a tracer for isoprene emissions (I agree), it should be acknowledged that organic aerosols can consist of up to thousands of different compounds.

Reply 32: At least the results indicate that the observed WSOC was more influenced by the emissions of  $\alpha$ -pinene or DMS rather than primary sea spray emissions based on the differences in each coefficient of determination. Taking account of the comment, we have modified the sentence as follows (L.289): "The overall results

suggest that the observed ocean-derived WSOC was affected by secondary formation from DMS-relevant compounds and α-pinene rather than primary sea spray emissions."

L262–263: "higher" and "lower" should be avoided. Were the differences in MSA and 3-MBTCA significant or not?

Reply 33: The differences in the MSA and 3-MBTCA concentrations were insignificant between the bloom and bloom-decay periods (p = 0.16 and 0.24, respectively). We modified the statement as follows:

L. 292: "The average MSA and 3-MBTCA concentrations during the bloom-decay period were similar to those during the bloom period (Table 3), where the differences are insignificant (p = 0.16 and 0.24 for MSA and 3-MBTCA, respectively)."

L264: The authors mention "increased sunlight intensity." Can the authors provide information about radiation in the area during the investigated period (e.g., from reanalysis products) to prove this claim?

Reply 34: The average direct solar radiation measured at Abashiri, a coastal city in the southern edge of the Sea of Okhotsk, showed that the intensity during the bloomdecay period (24.7 $\pm$ 14.4 MJ m-2) was indeed higher (p < 0.05) than that during the bloom period (11.1 $\pm$ 7.84 MJ m-2) (Japan Meteorological Agency). We believe that the difference in the solar radiation intensity between the two periods supports our measurement result. In the revised manuscript, this statement has been added to the text:

L.296: "Indeed, the average direct solar radiation measured at Abashiri, a coastal city in the southern edge of the Sea of Okhotsk, showed that the intensity during the bloomdecay period (24.7 $\pm$ 14.4 MJ m-2) was indeed higher (p < 0.05) than that during the bloom period (11.1 $\pm$ 7.84 MJ m-2) (Japan Meteorological Agency)."

L351–352: Could a correlation between WSON and Na help in discerning between direct emissions of N-containing compounds or secondary organic aerosol formation?

Reply 35: Yes, the R2 value between WSON and Na+ concentrations was below 0.01 (p = 0.05) during the bloom, as expected from the relationship between WSOC and Na+ concentrations. This suggests that the primary emission of sea spray made a minor contribution to WSON, supporting the large contribution of SOA to WSON in this study. We have made an additional description of this in the revised manuscript.

L.334: "In this study, the  $R^2$  value between WSON and  $Na^+$  concentrations in submicrometer aerosols was below 0.01 (p=0.05) during the bloom, as expected from the relationship between WSOC and  $Na^+$  concentrations. This also suggests the minor contribution of primary emission of sea spray to WSON, and the major contribution of SOA to WSON in the current study."

L365–366: Please provide statistics in support of this claim. To me, it appears that WSOC and WIOC are influenced by great internal variability, meaning that simply comparing averages can be misleading (unless a statistical test supports the claim).

Reply 36: In the revised manuscript, the p-value (p < 0.05) for the difference in the WSOM:OM and WIOM:OM has been provided to support the statement (L.254).

L368–370: Could you please provide the uncertainties associated with OM? Are these values significantly different?

Reply 37: The measurement uncertainty of WSOC was below 8% and the calculated uncertainty of WIOC was below 13%. Because we used empirical factors to derive WSOM and WIOM, the uncertainty of OM is larger than each uncertainty. As we wrote in the original text as well as in Reply 29, the difference in the OM:submicrometer mass ratio between the bloom and bloom-decay periods was insignificant (p = 0.23).

L370–372: I would water down this sentence. Considering that OA is composed of thousands of different compounds, I think it is too generalizing to attribute sources of WSOA only based on correlations with three chemical species.

Reply 38: At least a lack of correlation between WSOC and Na+ concentrations, together with the correlation with MSA and 3-MBTCA, suggests the larger

contributions of secondary formation to WSOC rather than primary emissions of sea spray aerosols. Taking into account the comment, we have modified the sentence as follows:

L.410: "Correlations between concentrations of WSOC and those of molecular tracers suggested that the ocean-derived WSOC observed in this study was likely affected by secondary formation rather than primary emissions of sea spray aerosols during the study period."

L373: "significantly lower": how did you justify this "significantly"?

Reply 39: In the revised manuscript, we added p < 0.05 for the difference in the WSOC:WSON ratios between the bloom and bloom-decay periods, together with their average values (L.320).

Table 1: I would also add values referring to WSON and C:N ratios.

Reply 40: The average values of WSON concentrations, WSOC:WSON, and DOC:DON ratios are added to Table 1 as suggested.

Table 2: "terretrial" is a typo.

Reply 41: It has been corrected.

Figure 3d: could you consider the possibility to add above -22 something like: "marine sources", and below -22 "terrestrial sources".

Reply 42: According to the suggestion, the terms "marine sources" and "terrestrial sources" have been added to Figure 2 with arrows to indicate their ranges.

**References**

- Miyazaki, Y., Yamashita, Y., Kawana, K., Tachibana, E., Kagami, S., Mochida, M., Suzuki, K., and Nishioka, J.: Chemical transfer of dissolved organic matter from surface seawater to sea spray water-soluble organic aerosol in the marine atmosphere, Scientific Reports, 8(1), 14861, https://doi.org/10.1038/s41598-018-32864-7, 2018.
- Miyazaki, Y., Suzuki, K., Tachibana, E., Yamashita, Y., Müller, A., Kawana, K., and Nishioka, J.: New index of organic mass enrichment in sea spray aerosols linked with senescent status in marine phytoplankton, Scientific Reports, 10(1), 17042, https://doi.org/10.1038/s41598-020-73718-5, 2020.

---

## Author Comment (AC3)

**Responses to the comments**

**We appreciate the additional comments on our work given by the editor and referee. Our responses to the specific comments and details of the changes made to the manuscript are given below.**

**The editor's comments and our responses**

In addition to the revisions requested by the anonymous reviewers, please address also the following requests.

In section 2.4.5:

・Add the list of all target analytes that were investigated in the study.

**Reply 1: Because the target compounds are only two (2-methyltetrols and 3-MBTCA) in this study and because they are already listed in Tables 3 and S2, we do not think it is necessary to additionally provide them in an individual list.**

・Provide full experimental details for the GC-Ms method used. This should include: chromatographic column used, elution method, amount injected, retention times of the analytes, quantification method, any use of calibration standards, method for determining recovery, method for determining reproducibility (or repeatability), limits of detection of the analytes.

**Reply 2: According to the comment, we have provided the details for the GC-MS experimental method, except for the retention time, as follows (L.165):**

*"…After the derivatization, the derivative was diluted with hexane containing the internal standard (n-Tridecane ($C_{13}$) with concentration of 1.43 µg ul$^{-1}$ in hexane). Two µL of the TMS derivative was then injected into a capillary gas chromatograph (GC8890, Agilent) equipped with a DB-5MS fused silica capillary column (30 m × 0.25 mm i.d., 0.25 µm film thickness) coupled to a mass spectrometer (MSD5977B, Agilent) to determine the concentration of each molecular tracer. The mass spectrometer was operated in the electron ionization (EI) mode at 70 eV. The sample injection was made in splitless mode. The peaks of the target compounds in total ion chromatograms (TICs) were identified by*

*comparison of mass spectra with those of authentic standards or literature data. 3-MBTCA was estimated using the response factor of pimelic acid, which was determined using an authentic standard. 2-methyltetrols was quantified using the response factor of meso-erythritol (Fu et al., 2009). The mass concentrations of molecular tracers were determined by the MS peak area of TMS derivative relative to that of the 140 µl internal standard injected into the GC-MS. Recoveries of each organic compound were measured using the surrogates that were spiked into precombusted quartz-fiber filters (n=3), which were higher than 81% for all the compounds measured. Reproducibility of the measurements is based on relative standard deviation of the concentrations based on duplicate analysis, which was generally <13%. …"*

**Retention time of a peak of a target compound depends on the length of the column in the GC instrument, which requires regular cutting during maintenance. The cutting includes removing short sections from the inlet end of the column to remove contaminants that are permanently retained on the column. Because the retention time changes depending on the length of the column, we do not include the information on it.**

Table S2: report results for all analytes investigated and report detection limits for all analytes reported in the table.

**Reply 3: According to the comment, all the aerosol chemical parameters investigated in this study are shown in Table S2 together with their lower detection limits of the concentrations for parameters directly measured.**

Please note typo at line 188 of the tracked-changes version: "chl.a".

**Reply 4: Corrected as pointed out (L.198).**

**Comments by the referee and our responses**

The authors have addressed almost all of my comments and I am content for the manuscript to be published. I would suggest a couple of changes that might be made.

Line 80 explain briefly the "blank procedure", since there are several ways to do this.

**Reply 1: Four field blanks were collected with quartz-fiber filters mounted on the impactor without running the HVAS, which were obtained on the ship during the expedition. This has been additionally described in the revised manuscript (L. 80).**

Line 223 I would prefer molar units, but if you use mass units please specify them more ng/m3 SO4 for example should specify if the weight is as S or SO4.

**Reply 2: The mass unit is used for sulfate, because the mass concentrations are compared with those of the other chemical component (i.e., OM, etc.) in this context. Also, as we clearly mention "Sulfate ($SO_4^{2-}$)" at the beginning of a sentence, we believe it is apparent that the concentration is for the sulfate mass rather than S. Therefore, we decided to keep them as they are.**

The paper usefully demonstrates that fine mode organic aerosol composition in this area is dominated by marine probably gaseous sources. I would however, suggest that the argument about the mechanism can be clarified a bit.

Firstly in line 220 the data seems to suggest that aerosol WSOC is similar before the bloom, so are the results really bloom period specific?

**Reply 3: First, the WSOC concentrations shown in Section 3.1 are those for all the data obtained during each period. Meanwhile, the difference in the WSOC concentrations of "marine origin" between the bloom (803±555 ngC m$^{-3}$; Table 1) and the pre-bloom (545±332 ngC m$^{-3}$) periods is larger than that for all the data. In addition, the contribution of sea spray aerosols to WSOC was more significant during the pre-bloom period (Miyazaki et al., 2018) compared to the bloom period in this study. Therefore, these results support our conclusion that aerosol WSOC and WSON of marine origin were likely affected by secondary formation from precursors of marine origin rather than primary emissions of sea spray aerosols.**

Secondly there seems some contradiction in the suggested sources of some of the WSOC and WSON in aerosols.

In lines 325 and 370 there is discussion of proteins in seawater, but I assume the authors are not suggesting that these are volatile.

The discussion around line 335 links the seawater DOC and DON to the aerosol WSOC and WSON, but most of the DOC and DON in seawater is not volatile – indeed we know it is high molecular weight and recalcitrant- and the authors acknowledged in the response to reviewers that amines while volatile form seawater are at very low concentrations.

I would suggest the authors clarify (or remove) the mechanisms they are proposing

**Reply 4: We do not intend to mention that the seawater DOC and DON compositions were preserved during their sea-to-air emissions, but some parts of them may have been affected by photodegradation and/or biodegradation in the air-sea interface to produce more volatile compounds. Although it is difficult to provide a clear explanation of the exact mechanism for the aerosol WSON formation including the processes in the air-sea interface, we have added the following statement in the revised manuscript, taking account of the referee's comment:**

**L. 407: *"It is noted that the majority of DOC and DON discussed in this study are generally high molecular weight compounds and have low volatility. Therefore, photodegradation and/or biodegradation of DOC and DON in the air-sea interface are likely important to produce more volatile compounds for the atmospheric emissions, which needs further investigation in future studies."***

Reference

Miyazaki, Y., Yamashita, Y., Kawana, K., Tachibana, E., Kagami, S., Mochida, M., Suzuki, K., and Nishioka, J.: Chemical transfer of dissolved organic matter from surface seawater to sea spray water-soluble organic aerosol in the marine atmosphere, Scientific Reports, 8(1), 14861, https://doi.org/10.1038/s41598-018-32864-7, 2018.

---

## Author Response (AR3)

**Responses to the comments**

**We appreciate the editor's additional comment on the manuscript. Our response to the comment and the change made to the manuscript (Supplement) are given below.**

**The editor's comment and our response**

From the pre-print manuscript and answers to the reviewers, it is clear that 3-MBTCA and 2-methyltetrols were not the only organic compounds being determined. Glucose, pinic acid and pinonic acid were also mentioned. Please add those to Table S2 for completeness. Would any discussion point or conclusion need to be changed as a result of this requested addition?

**Reply: According to the comment, the data of the three organic compounds (glucose, pinic acid, and pinonic acid) have been added to Table S2 in the Supplement. This revision will not affect any points or conclusions in the manuscript.**

**Comments by the referee and our responses**

The authors have addressed almost all of my comments and I am content for the manuscript to be published. I would suggest a couple of changes that might be made.

Line 80 explain briefly the "blank procedure", since there are several ways to do this.

**Reply 1: Four field blanks were collected with quartz-fiber filters mounted on the impactor without running the HVAS, which were obtained on the ship during the expedition. This has been additionally described in the revised manuscript (L. 80).**

Line 223 I would prefer molar units, but if you use mass units please specify them more ng/m3 SO4 for example should specify if the weight is as S or SO4.

**Reply 2: The mass unit is used for sulfate, because the mass concentrations are compared with those of the other chemical component (i.e., OM, etc.) in this context. Also, as we clearly mention "Sulfate ($SO_4^{2-}$)" at the beginning of a sentence, we believe it is apparent that the concentration is for the sulfate mass rather than S. Therefore, we decided to keep them as they are.**

The paper usefully demonstrates that fine mode organic aerosol composition in this area is dominated by marine probably gaseous sources. I would however, suggest that the argument about the mechanism can be clarified a bit.

Firstly in line 220 the data seems to suggest that aerosol WSOC is similar before the bloom, so are the results really bloom period specific?

**Reply 3: First, the WSOC concentrations shown in Section 3.1 are those for all the data obtained during each period. Meanwhile, the difference in the WSOC concentrations of "marine origin" between the bloom (803±555 ngC m$^{-3}$; Table 1) and the pre-bloom (545±332 ngC m$^{-3}$) periods is larger than that for all the data. In addition, the contribution of sea spray aerosols to WSOC was more significant during the pre-bloom period (Miyazaki et al., 2018) compared to the bloom period in this study. Therefore, these results support our conclusion that aerosol WSOC and WSON of marine origin were likely affected by secondary formation from precursors of marine origin rather than primary emissions of sea spray aerosols.**

Secondly there seems some contradiction in the suggested sources of some of the WSOC and WSON in aerosols.

In lines 325 and 370 there is discussion of proteins in seawater, but I assume the authors are not suggesting that these are volatile.

The discussion around line 335 links the seawater DOC and DON to the aerosol WSOC and WSON, but most of the DOC and DON in seawater is not volatile – indeed we know it is high molecular weight and recalcitrant- and the authors acknowledged in the response to reviewers that amines while volatile form seawater are at very low concentrations.

I would suggest the authors clarify (or remove) the mechanisms they are proposing

**Reply 4: We do not intend to mention that the seawater DOC and DON compositions were preserved during their sea-to-air emissions, but some parts of them may have been affected by photodegradation and/or biodegradation in the air-sea interface to produce more volatile compounds. Although it is difficult to provide a clear explanation of the exact mechanism for the aerosol WSON formation including the processes in the air-sea interface, we have added the following statement in the revised manuscript, taking account of the referee's comment:**

**L. 407:** *"It is noted that the majority of DOC and DON discussed in this study are generally high molecular weight compounds and have low volatility. Therefore, photodegradation and/or biodegradation of DOC and DON in the air-sea interface are likely important to produce more volatile compounds for the atmospheric emissions, which needs further investigation in future studies."*

Reference

Miyazaki, Y., Yamashita, Y., Kawana, K., Tachibana, E., Kagami, S., Mochida, M., Suzuki, K., and Nishioka, J.: Chemical transfer of dissolved organic matter from surface seawater to sea spray water-soluble organic aerosol in the marine atmosphere, Scientific Reports, 8(1), 14861, https://doi.org/10.1038/s41598-018-32864-7, 2018.